# Genome Sequencing of the Japanese Eel (*Anguilla japonica*) for Comparative Genomic Studies on *tbx4* and a *tbx4* Gene Cluster in Teleost Fishes

**DOI:** 10.3390/md17070426

**Published:** 2019-07-20

**Authors:** Weiwei Chen, Chao Bian, Xinxin You, Jia Li, Lizhen Ye, Zhengyong Wen, Yunyun Lv, Xinhui Zhang, Junmin Xu, Shaosen Yang, Ruobo Gu, Xueqiang Lin, Qiong Shi

**Affiliations:** 1BGI Education Center, University of Chinese Academy of Sciences, Shenzhen 518083, China; 2Shenzhen Key Lab of Marine Genomics, Guangdong Provincial Key Lab of Molecular Breeding in Marine Economic Animals, BGI Academy of Marine Sciences, BGI Marine, BGI, Shenzhen 518083, China; 3BGI Zhenjiang Institute of Hydrobiology, Zhenjiang 212000, China; 4Guangdong Provincial Key Laboratory for Aquatic Economic Animals, School of Life Sciences, Sun Yat-Sen University, Guangzhou 510275, China

**Keywords:** Japanese eel (*Anguilla japonica*), genome sequencing and assembly, *tbx4*, *tbx4* gene cluster, pelvic fin, fin spine, teleost fish

## Abstract

Limbs originated from paired fish fins are an important innovation in Gnathostomata. Many studies have focused on limb development-related genes, of which the T-box transcription factor 4 gene (*tbx4*) has been considered as one of the most essential factors in the regulation of the hindlimb development. We previously confirmed pelvic fin loss in *tbx4*-knockout zebrafish. Here, we report a high-quality genome assembly of the Japanese eel (*Anguilla japonica*), which is an economically important fish without pelvic fins. The assembled genome is 1.13 Gb in size, with a scaffold N50 of 1.03 Mb. In addition, we collected 24 *tbx4* sequences from 22 teleost fishes to explore the correlation between *tbx4* and pelvic fin evolution. However, we observed complete exon structures of *tbx4* in several pelvic-fin-loss species such as Ocean sunfish (*Mola mola*) and ricefield eel (*Monopterus albus*). More interestingly, an inversion of a special *tbx4* gene cluster (*brip1*-*tbx4*-*tbx2b*- *bcas3*) occurred twice independently, which coincides with the presence of fin spines. A nonsynonymous mutation (M82L) was identified in the nuclear localization sequence (NLS) of the Japanese eel *tbx4*. We also examined variation and loss of hindlimb enhancer B (HLEB), which may account for pelvic fin loss in Tetraodontidae and Diodontidae. In summary, we generated a genome assembly of the Japanese eel, which provides a valuable genomic resource to study the evolution of fish *tbx4* and helps elucidate the mechanism of pelvic fin loss in teleost fishes. Our comparative genomic studies, revealed for the first time a potential correlation between the *tbx4* gene cluster and the evolutionary development of toxic fin spines. Because fin spines in teleosts are usually venoms, this *tbx4* gene cluster may facilitate the genetic engineering of toxin-related marine drugs.

## 1. Introduction

The Japanese eel, *Anguilla japonica*, is a world-famous teleost fish due to its unique migration pattern and economic importance. Without any effective breeding technology, aquaculture in Asian countries has to depend on catching wild glass eels each year. To provide genetic resources for biological and practical studies on this teleost, we initiated a Japanese eel genome project in China.

The emergence of paired appendages has improved the movability and defensive capability of ancient vertebrates. In fishes, pectoral fins first appeared in extinct jawless fishes, whereas pelvic fins initially developed in the most primitive extinct jawed fishes—placoderms—which existed ~525 million years ago (Mya) in the middle Cambrian [1,2,3,4,5]. Tetrapods evolved from a fish-like ancestor; subsequently, paired fins evolved into limbs to adapt to terrestrial environments. Hence, it was a common thought that tetrapod forelimbs and hindlimbs are homologous to fish pectoral and pelvic fins, respectively. In recent years, the evolution of paired appendages has drawn considerable attention. Similar to the Japanese eel, the tiger tail seahorse (*Hippocampus comes*) exhibits a pelvic- fin-loss phenotype, which may be due to the loss of the T-box transcription factor 4 (*tbx4*), as we have confirmed pelvic fin loss in *tbx4*-knockout zebrafish [6].

T-box genes encode a family of transcription factors that are related to the metazoan development [7]. Their protein sequences possess a highly conserved DNA-binding domain (i.e., T-box domain). Within the T-box family, *tbx2/3/4/5* subfamilies have been extensively studied because of their important roles in development of vertebrate appendages, heart and eyes [8]. At least 600 Mya, *tbx2/3* and *tbx4/5* diverged via tandem duplication, which maintained a tight linkage in most species; subsequently, either of these duplicated into two paralogous genes due to a whole genome duplication event [8]. *Tbx4* has been a principal effective gene for the development of pelvic fins or hindlimbs in vertebrates, as knocking out *tbx4* in zebrafish disrupts pelvic fin formation [6]. To date, numerous studies on *tbx4* have mainly focused on mammals, especially on humans, whereas those involving fishes are limited. With over 38,000 extant species, teleost fishes comprise the largest group of living vertebrates [9] for in-depth investigations on the evolution of the *tbx4* gene.

The development of high-throughput sequencing technologies has facilitated sequencing of the genomes of over 60 teleost species [10]. These data provide a good opportunity for us to perform a comparative genomic study on *tbx4* isotypes and to identify variations in *tbx4* sequences and gene structures across teleost fishes. In this study, we generated a high-quality genome assembly of the Japanese eel and then performed phylogenetic and synteny analyses, as well as variation determination of 24 *tbx4* sequences in 22 representative teleost species. Interestingly, for the first time, we determined an inversion of a special *tbx4* gene cluster that is potentially correlated with the evolutionary development of teleost fin spines, which may facilitate the development of marine drugs.

## 2. Results

### 2.1. Summary of Genome Survey and De Novo Assembly

A total of 268.61 Gb of raw reads were generated by an Illumina HiSeq X-Ten platform (see more details in Section 4.2 and Appendix A). After removal of low-quality reads, index and adapter sequences, and PCR duplicates using SOAPfilter v2.2 (BGI, Shenzhen, China) [11], we obtained 184.05 Gb of clean reads (Appendix A) for subsequent assembly and annotation. Based on a 17-mer distribution (Figure 1), we estimated that the genome size of the Japanese eel is approximately 1.03 Gb (Appendix A) [12].

We employed SOAPdenovo [11] to obtain a primary draft, which consisted of 1,227,464 contigs and 462,272 scaffolds, with a contig N50 of 2.00 kb and a scaffold N50 of 383.80 kb. Subsequently, GapCloser and SSPACE [13] were employed to fill the gaps and elongate the scaffolds. As a result, we assembled a 1.23-Gb genome with 608,352 contigs and 351,879 scaffolds (Table 1). To assess the completeness of this assembly, we employed actinopterygii_bod9 as reference for a BUSCO analysis [14,15], which demonstrated that the benchmarking value was fair up to 83.9% (Table 2). 

After performing an additional series of filtering manually for those heterozygous redundant scaffolds (sequencing depth <40×; see more information in Section 4.2), we generated a final 1.13-GB genome assembly (Table 1) with 256,649 contigs and 41,687 scaffolds, and the contig N50 and scaffold N50 values reached 11.47 kb and 1.03 Mb, respectively.

In our assembly, repetitive sequences accounted for 22.94% of the entire genome. The detailed categories are summarized in Appendix A. Finally, a total of 17,147 genes with an average of 7.6 exons were predicted (see more details in Appendix A).

### 2.2. Conservation of the Vertebrate tbx4 Genes in Gene Structure

The examined various vertebrate species, including amphioxus (*Branchiostoma floridae*), a shark (*Chiloscyllium punctatum*), the zebrafish (*Danio rerio*), a frog (*Xenopus tropicalis*), a turtle (*Chelonia mydas*), a chicken (*Gallus gallus*) and humans (*Homo sapiens*), were chosen as representative of Cephalochordata, cartilaginous fishes, bony fishes, amphibians, reptiles, and mammals, respectively (see more details in Section 4.3 and Figure 2). From amphioxus to human, we observed that the *tbx4* genes in all vertebrate species contain eight exons, and each of them have a similar length in various species (Figure 2). Although divergent at ~699 Mya [16], this gene seems to be highly conserved in terms of gene structure in vertebrates.

### 2.3. Conservation of the T-box Region

All T-box genes share a common T-box domain, which is composed of 180 to 190 amino acid (aa) residues [17]. Multiple sequence alignment of 24 *tbx4* sequences from 22 representative species revealed that the T-box domains of the *tbx4* genes are highly conserved in vertebrates (Figure 3). Usually, the central T-box domain is composed of exons 3 to 5, partial exon 2 and exon 6. Interestingly, a cave-restricted barbel fish (*Sinocyclocheilus anshuiensis*) [18] and anadromous Chinook salmon (*Oncorhynchus tshawytscha*) [19] both possess two copies of the *tbx4* gene, most likely due to an extra whole genome duplication event.

### 2.4. The Tbx4 Gene of the Japanese Eel

The nuclear localization sequence (NLS) of *tbx4* genes usually consists of 13 aa and lies in the conserved DNA-binding motif (T-box domain; Figure 3). Comparative analysis revealed a nonsynonymous variant in the Japanese eel *tbx4* (M82L; Figure 4), which may be related to pelvic fin loss [20]. To confirm this variant, PCR was performed (data not shown).

A previous research [21] showed that all substitutions except for K11 of *tbx5* NLS could cause cytoplasmic localization of fusion proteins. Another study [20] identified two nonsynonymous mutations in the NLS of zebrafish *tbx4*, which impaired nuclear localization of the protein and thereby disrupted pelvic fin development.

The entire protein sequence of the Japanese eel *tbx4* gene is presented in Figure 5. Its alignment with zebrafish TBX4 revealed a high conservation between the two fish species. Localization of the same NLS sequence in both fishes is marked in a red box for a detailed comparison (upper panel in Figure 5).

### 2.5. Phylogenetic Analysis and Synteny Comparison

We constructed a phylogenetic tree of 24 *tbx4* protein sequences using spotted gar (*Lepisosteus oculatus*) as the outgroup (Figure 6). Eels (Elopomorpha) showed the earliest branching in teleosts, whereas the other lineages diverged later. Because the length of each branch represents the evolution rate of the examined gene, we speculate that the Japanese eel has a high evolution rate in *tbx4* (see more details in the left tree of Figure 6).

### 2.6. The Brip1-tbx4-tbx2b-bcas3 Cluster

We observed that a cluster composed of four genes (*brip1*, *tbx4*, *tbx2b* and *bcas3*) maintains the same arrangement in teleosts (Figure 6 and Figure 7), despite chromosomal rearrangements occurring since the divergence of bony vertebrates approximately 465 Mya [22,23,24,25]. Previous studies involving mammals demonstrate a putative limb enhancer at the interior of *bcas3* [26]. The hindlimb enhancer A (HLEA) and HLEB are located at the interval of *tbx2-tbx4* and *tbx4*-*brip1*, respectively [27]. *Tbx2* encodes a transcriptional repressor that is related to digit development [28]. *Bcas3* has shown to be overexpressed in breast cancer cells [29], and *brip1* encodes a protein that interacts with *bcas3* [30]. However, current understanding of *brip1* and *bcas3* in fishes is limited.

Interestingly, an inversion of the *brip1*-*tbx4*-*tbx2b*-*bcas3* cluster occurred twice independently in teleosts. One inversion happened in Acanthopterygii, and the another appeared in a subclade of Otophysa—which includes Characiformes, Gymnotiformes and Siluriformes—but not Cypriniformes (see more details in the two highlighted black boxes of Figure 6). Lineages with one of the inversions are armed with fin spines. In fact, Acanthopterygii is named for the representative sharp and bony rays in their dorsal fins, anal fins or pelvic fins. Members of Siluriformes are armed with spines in the anal, dorsal, caudal, adipose, and paired fins [31,32,33]. Additionally, fin spines are characterized in some members of the Characiformes [34]. Gymnotiformes, although divergent from Siluriformes [35], is an outlier due to its absence of pelvic fins and dorsal fins.

Chinese yellow catfish (*Pelteobagrus fulvidraco*) secretes venom through its fin spines, which has been proposed by us to be essential for the development of marine drugs [36]. Eeltail catfish (Siluriformes), scorpionfish and stonefish (Scorpaeniformes) also have venomous fin spines that can severely injure other animals [37]. As we determined in the present study, an inversion of the *brip1-tbx4-tbx2b-bcas3* cluster occurred in these Acanthopterygii fishes (Figure 7), which are in line with the existence of fin spines.

### 2.7. HLEB

HLEB is a highly conserved enhancer of the *tbx4* genes from mammals to cartilaginous fishes, which play an important role in hindlimb development [27]. Here, we compared 11 HLEB sequences across Acanthopterygii fishes against three-spined stickleback (*Gasterosteus aculeatu*). Previous studies suggested that Tetraodontiformes had undergone reductions or increases in pelvic complexes [38]. We found that the HLEB of Ocean sunfish (*Mola mola*) was very similar to three-spined stickleback than other four related puffer fishes (*Takifugu bimaculatus*, *T. obscurus*, *T. rubripes*, and *Tetraodon nigroviridi*; see a VISTA plot [39,40] in Figure 8), which might be responsible for the loss of pelvic fins in Tetraodontidae. However, another Tetraodontiformes species, spot-fin porcupinefish (*Diodon hystrix*) as well as tiger tail seahorse may have lost the HLEB sequence (unpublished data).

## 3. Discussion

### 3.1. Various Genetic Mechanisms for Pelvic Fin Development

Since the emergence of two paired appendages, one or both of these were secondarily lost in many animal lineages and showed a corresponding high level of disparity. For example, eels (Anguilliformes), ricefield eel (Synbranchiformes), and electric eel (*Electrophorus electricus*; Gymnotiformes) have completely lost their pelvic fins completely; for puffer fishes and filefishes (Tetraodontiformes), however, there exists a great diversity in their pelvic fins ranging from acquired to various degrees of reduction [38]. It has been proposed that an altered *hoxd9a* expression may account for the loss of pelvic fins in Japanese puffer (*Takifugu rubripes*; Tetraodontiformes) [41]. Basal snakes (boa and python) retained a vestigial pelvic girdle and rudimentary hindlimbs, whereas advanced snakes (viper, rattlesnake, king cobra, and corn snake), representing the majority (>85%) of all extant snake species, completely lost all skeletal limb structures due to a 17-bp deletion in the zone of the polarizing activity (ZPA) regulatory sequence [42]. The ZPA has proven to be a limb-specific enhancer of the Sonic hedgehog (*Shh*) gene, which is indispensable for limb development [43,44,45,46,47,48,49,50]. In addition, another research demonstrated that the HLEB, a highly conserved putative *pitx1* binding site, had lost the function for limb development in snakes [27].

Two nonsynonymous mutations within the *tbx4* NLS (A78V, G79A) are enough to disrupt pelvic fin development in zebrafish [20]. *Pitx1*, a homeobox-containing transcription factor with importance in hindlimb identity and outgrowth [51,52,53], has been associated with pelvic fin variations in natural populations of three-spined stickleback (Gasterosteiformes) [54]. *Pitx1*–mediated pelvic reduction was also observed in ninespine stickleback (*Pungitius pungitius*; Gasterosteiformes), and even in distantly related species such as manatees [54]. Furthermore, some lizards and mammals have more or less lost their paired appendages, although the detailed mechanisms remain unclear.

Embryonic development of limbs or pelvic fins mainly undergoes three main steps, including positioning, initiation and outgrowth. It is a comprehensive process and involves several genes, such as *tbx4*, *pitx1*, *hoxa13*, *hoxb9*, *hoxc9*, *hoxd9*, *hoxd10*, *hoxd13*, *wnt2b*, *wnt8c*, *wnt3a/3*, *shh*, *fgf10*, and *fgf8* [41,42,43,44,45,46,47,48,49,50,51,52,53,54,55]. At the first step of embryonic development of vertebrate paired appendages, homeobox (*hox*) genes expressed in the lateral plate mesoderm and somitic mesoderm specify the position of the limbs and interlimb region. Subsequently, *tbx4* and *tbx5* in the lateral plate mesoderm activate the expression of downstream *wnt8c/fgf10* and *wnt2b/fgf10* in hindlimbs and forelimbs, respectively. Then, Wnt/Fgf signaling feedbacks on *tbx4* and *tbx5* to maintain their expression. After that, *fgf10* activates *wnt3a/3/fgf8* signals in the limb ectoderm and induces the formation of apical ectodermal ridge *shh* expression in the posterior limb bud, which has been maintained by fibroblast growth factor (FGF) signaling in the apical ectodermal ridge and the FGF signaling feedback on *shh* to maintain its expression. In addition, *tbx4* expression is regulated by *pitx1* and to a lesser extent by *pitx2* [53].

Here, we selected one of the most important factors for the involvement in hindlimb development—*tbx4*—to perform comparative genomic studies in teleost fishes. We provided new information about *tbx4* genes, including gene structure, variation, synteny, enhancement sequence and phylogenetic status. We also revealed that the genetic backgrounds for pelvic fin loss in various species might be diverse. *Monopterus albus* has lost pelvic fins despite a complete gene structure for *tbx4* and normal HLEB. The genetic mechanisms of pelvic fin loss in a given group of species, such as pufferfishes, may significantly vary. Previous studies have suggested that altered *hoxd9a* expression may result in pelvic loss in Japanese puffer [41]. According to our present results, however, variations in HLEB may be also involved in the regulation of pelvic phenotypes.

### 3.2. Potential Importance of the Tbx4 Gene Cluster for the Evolutionary Development of Toxic Fin Spines

Structural conservation often indicates stable function(s). We observed that the *tbx4* gene cluster *brip1*-*tbx4-tbx2b-bcas3* widely exists in teleost fishes. Previous studies have demonstrated that this cluster linkage may result from the shared regulatory domains required for coordinated expression [8]. The NLS of *tbx4* plays a key role in protein nuclear transport, and its structure must be intact to play its essential role in the induction of the pelvic fin outgrowth [21]. For the Japanese eel, a nonsynonymous mutation was detected in the NLS of *tbx4* (Figure 4), which is considered to be correlated with pelvic fin development.

Many fishes have developed fin spines for defense or hunting purposes. In our previous reports [36,56,57], we predicted several toxin genes from the venom glands of fin spines in Chinese yellow catfish using a combination of genomic, transcriptomic, and proteomic sequencing. The contribution of more toxins to future drugs seems to be more promising [57], and we will therefore sequence and analyze more fish species with fin spines. In Figure 6, we observed an inversion of the *tbx4* gene cluster, which may be correlated with the development of toxic fin spines. Hence, we propose a deep investigation of the synthetic biological application of this cluster, which may benefit the development of novel marine drugs.

## 4. Materials and Methods

### 4.1. Sample Collection

A female Japanese eel was collected from a local aquaculture base of BGI Marine in Huizhou, Guangdong Province, China. Species identification with cloning of the *COI* sequence was performed immediately after collection of muscle samples. All experiments were conducted in accordance with the guidelines of the Animal Ethics Committee and were approved by the Institutional Review Board on Bioethics and Biosafety of BGI (No. FT1510).

### 4.2. Genome Sequencing, Assembling and Annotation

Genomic DNA (for the genome sequencing) and total RNA (for the transcriptome sequencing) were extracted from the muscle samples as previously described [36]. For whole-genome sequencing, we constructed seven paired-end sequencing libraries, including three short-insert (270, 500, and 800 bp) and four long-insert (2, 5, 10, and 20 kb). Finally, paired-end sequencing was performed on an Illumina HiSeq X-ten platform (San Diego, CA, USA).

After genome sequencing, we employed the SOAPdenovo (version 2.04) to assemble the draft genome with the parameter “-k 27 –M 1”. Subsequently, krskgf, Gapcloser1.12, and Gapcloser1.10 were used to fill gaps in the primary assembly successively. After that, SSPACE was used to elongate the scaffolds produced by Gapcloser1.10. These steps were described in detail in our previous studies [12,18,36]. We also manually filtered those redundant scaffolds caused by the high heterozygosity. Because a heterozygous scaffold contains remarkably lower sequencing depth than a normal scaffold, we could remove these scaffolds with the sequencing depth <40× (~1/4 of the average sequencing depth). Our genome assembly of the Japanese eel has been deposited in the NCBI under the project ID PRJNA533944 with an accession code of VDMF00000000.

We used RepeatModeller v1.08 (Institute for System Biology, Seattle, CA, USA) along with LTR-FINDER v1.06 [58] for de novo repeat sequence prediction, and Tandem Repeat Finder (Trf), RepeatMasker v4.06 [59] along with RepeatProteinMask v4.06 for homology prediction by aligning to the RepBase v21.01 [60]. Finally, we integrated the results produced by the above-mentioned two prediction methods.

For whole gene set annotation, we masked the repetitive elements of the assembling genome and then adopted three different strategies, namely, ab initio annotation, homologous annotation, and transcriptome-based annotation, as previously reported [36]. We used AUGUSTUS v2.5 [61] and GENSCAN v1.0 [62] for ab initio prediction. For homologous annotation, we downloaded the protein sequences (release version 89) of eight vertebrate species from the ensemble, including zebrafish, Atlantic cod (*Gadus morhua*), spotted gar (*Lepisosteus oculatus*), Nile tilapia (*Oreochromis niloticus*), medaka (*Oryzias latipes*), Japanese puffer, spotted green pufferfish (*Tetraodon nigroviridis*), and sea lamprey (*Petromyzon marinus*) to search for the best-hit alignments in the Japanese eel genome by TblastN program [63]. Subsequently, GeneWise v2.2.0 [64] was used to identify the gene structure of alignment produced by TblastN. For transcriptome-based prediction, we used Tophat v2.1.1 [65] and Cufflinks v2.2.1 (University of Maryland, College Park, MD, USA) to predict the gene set with the transcriptomic data of liver and gill sequenced by an Illumina Hiseq2500 platform. Finally, EVidenceModeler [66] was employed to integrate the consensus results of the three prediction methods. The predicted gene set was used to identify the functional motifs and domains by mapping to five public functional databases, including KEGG [67], NCBI-Nr, Swiss-Prot, TrEMBL [68], and Interpro [69] using BLAST.

### 4.3. Collection of the Genome Sequences

We downloaded 26 fish genomes and 27 protein sequences of T-box family as well as seven adjacent proteins of *tbx4* from NCBI (see more details in Table 3). The Chinese clearhead icefish (*Protosalanx hyalocranius*) and Northern snakehead (*Channa argus*) genomes were downloaded from GigaDB [70,71]. The genomes of spot-fin porcupinefish (*Diodon hystrix*) and river fugu (*Takifugu obscurus*) were obtained from our laboratory (unpublished data).

### 4.4. Collection of the Tbx4 Sequences

We picked out seven genes with adjacent locations to the *tbx4* gene(s) in most of the downloaded fish genomes. To avoid mapping onto other paralogous genes in the T-box family and to ensure the accuracy to find *tbx4* homolog, we merged the 27 T-box protein sequences and seven adjacent genes (see Table 3) as a whole reference to build an alignment index. Subsequently, we aligned the reference to all the examined genomes using TBLASTN to acquire *tbx4* homolog sequence(s). We then selected those with alignments to at least three adjacent genes on the same chromosome or scaffold (with *tbx4* gene). Subsequently, we used Exonerate [70] and GeneWise v2.2.0 [62] to calculate the amino acid sequence of each *tbx4* gene, and we corrected errors manually according to the zebrafish TBX4 protein sequence. Finally, we obtained 24 TBX4 protein sequences from 22 representative teleost fishes.

Due to the limitations of sequencing and assembly, the Japanese eel *tbx4* sequence of our assembly was truncated. We filled the gap and completed the synteny information using a chromosome-level assembly version of the Japanese eel genome, GCA_003597225.1, from NCBI [72].

### 4.5. Sequence Alignment, Phylogenetic Analysis and Identification of Conserved Synteny

We extracted the T-box domain of these *tbx4* proteins and performed a multiple alignment by Muscle [73,74]. After that, we colorized the alignment results using TEXshade [75]. These collected TBX4 protein sequences were then employed to predict their best nucleotide substitution model under the Akaike Information Criterion (AIC) [76], which was implemented in prottest-3.4.2 [77]. We also performed multiple alignments of these collected *tbx4* protein sequences by MEGA-7.0 [78] and constructed phylogenetic topologies with 1,000 replicates to evaluate branch supports with the maximum likelihood (ML) method by phyML-3.1 [79,80]. To assess the collinearity conservation and assure confidence of the collected *tbx4* sequences, we detected arrangement orders of the seven adjacent genes of *tbx4* in each species.

### 4.6. HLEB Analysis

We obtained the 873-bp HLEB sequence of the three-spined stickleback by examining the reported primers on the genome (GCA_000180675.1). Then, we mapped this HLEB sequence onto the examined genomes for acquisition of corresponding homologous sequences by using LAGAN [81]. The alignment results were visualized by VISTA plot [39,40].

## 5. Conclusions

We sequenced and assembled a 1.13-Gb genome of the Japanese eel for a comparative genomic study on the *tbx4* gene cluster. The *tbx4* gene apparently harbors a nonsynonymous mutation in an important site of the NLS, which was considered to be correlated with the pelvic fin development. Interestingly, its adjacent *brip1* gene was also lost. We investigated 24 *tbx4* sequences from 22 teleost lineages and detected an inversion that occurred twice independently in teleost fishes, which coincides with the presence of fin spines. Additionally, the change or loss of HLEB may be responsible for the disappearance of pelvic fins in some Tetraodontiformes species. This is the first report describing the potential correlation of the inversed *tbx4* gene cluster with the development of fin spines, which may benefit the development of novel marine drugs.

## Figures and Tables

**Figure 1 marinedrugs-17-00426-f001:**
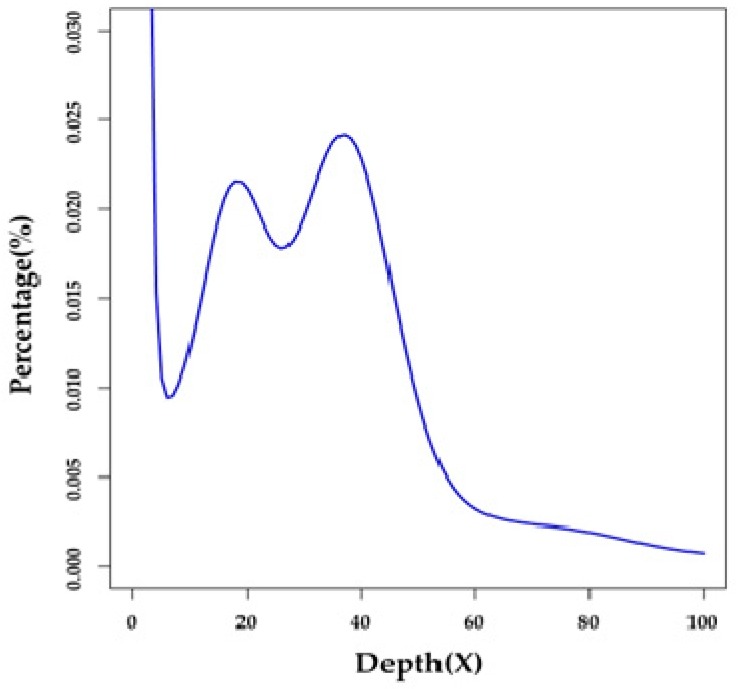
A 17-mer distribution of the Japanese eel genome sequencing. Only the sequencing data from short-insert libraries (500 and 800 bp) were used for the k-mer analysis. The x-axis is the sequencing depth of each unique 17-mer, and the y-axis is the percentage of these unique 17-mers. The peak depth (K_depth) is 37, and the corresponding k-mer number (N) is 37,982,773,125. We therefore calculated the genome size (G) to be ~1.03 Gb based on the following formula [12]: G=N/K_depth.

**Figure 2 marinedrugs-17-00426-f002:**
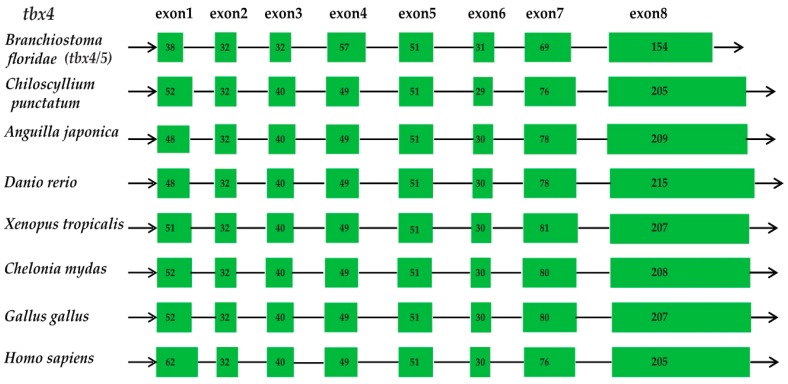
Structures of the *tbx4* genes in various vertebrate species. Green boxes and lines represent exons and introns, respectively. Numbers inside the boxes are the exact amino acid numbers, indicating their similarity among various species.

**Figure 3 marinedrugs-17-00426-f003:**
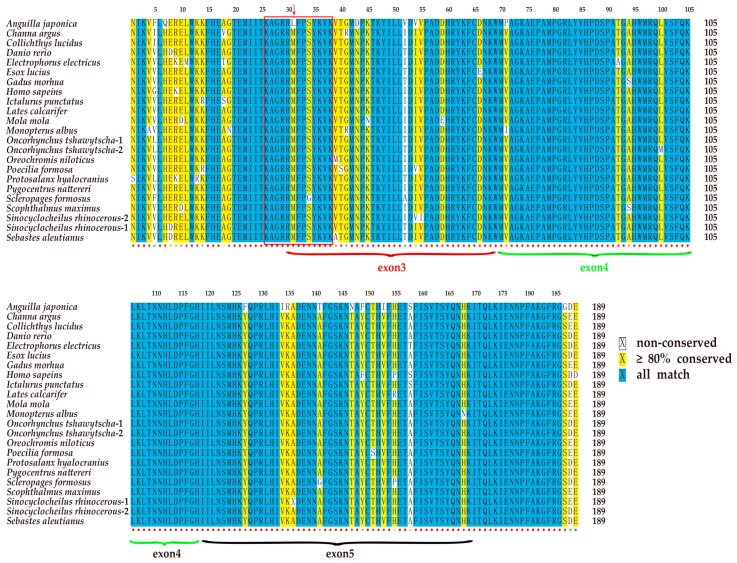
Similarity of the T-box domain in the *tbx4* genes of various vertebrate species. The blue color represents conserved sites. The yellow color represents the similarity >80% and white shows less conserved sites. The colored dots beneath the sequence alignment indicate conservation track, ranging from blue (non-conserved) to red (the most conserved). The red box highlights the NLS region of the *tbx4* genes. A red arrow indicates a nonsynonymous mutation in the Japanese eel *tbx4* gene.

**Figure 4 marinedrugs-17-00426-f004:**
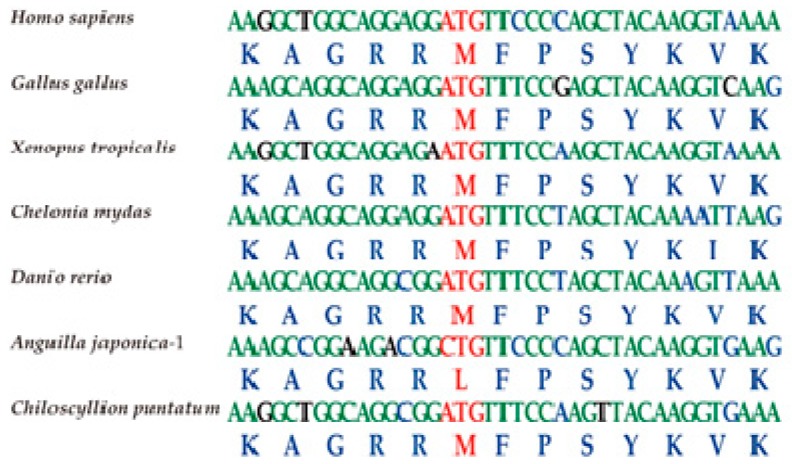
Visualization of the *tbx4* NLS regions in different vertebrate species using Bioedit (Tom Hall Ibis Therapeutics, Carlsbad, CA, USA). Red characters represent the codons and amino acids of nonsynonymous mutations.

**Figure 5 marinedrugs-17-00426-f005:**
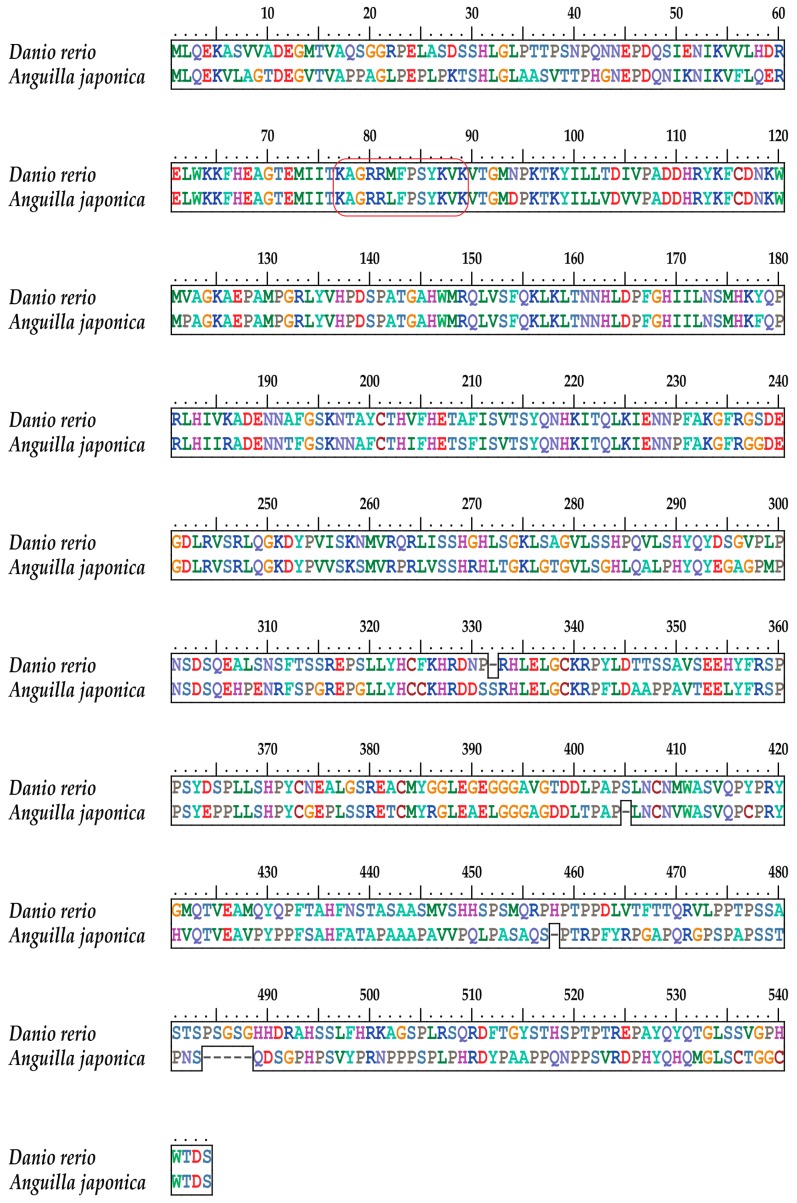
Alignment of the TBX4 protein sequences of Japanese eel against zebrafish. The red box highlights the NLS regions (same as Figure 4).

**Figure 6 marinedrugs-17-00426-f006:**
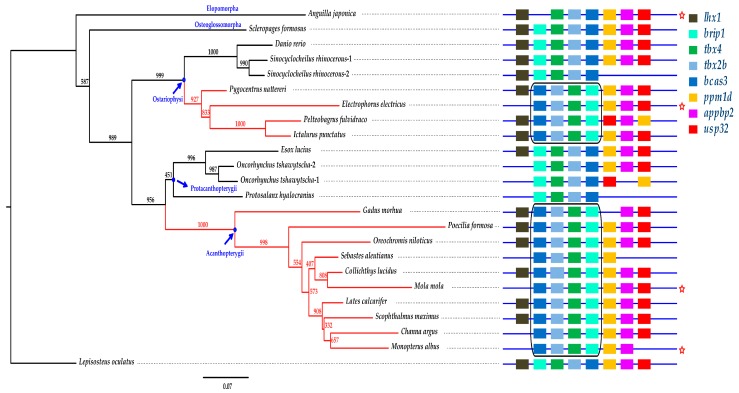
Phylogenetic and synteny comparisons of the *tbx4* genes in vertebrates. The figure in the left is a Bayesian tree. Numbers on the branches are bootstrap supports (black) obtained from the phyML-3.1 reconstruction. Spotted gar was used as the outgroup. The figure in the right represents the synteny of *tbx4*. Distances between genes and the gene length are not drawn to scale. The red branches and the two black boxes highlight two remarkable inversions of the *brip1*-*tbx4*-*tbx2b*-*bcas3* cluster in teleost species. The five-point star represents the determined pelvic fin loss in the examined species.

**Figure 7 marinedrugs-17-00426-f007:**
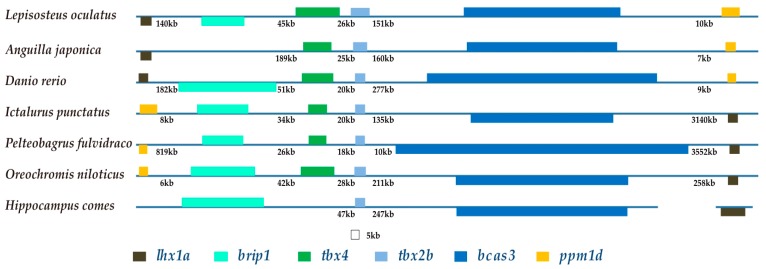
The *brip1-tbx4-tbx2b-bcas3* cluster. Colored boxes and lines represent genes and intergenic regions, respectively. The distance between two adjacent genes is indicated underneath the lines, while the length of exons is drawn to scale. Genes in the same orientation as *tbx4* are marked above the horizontal lines; however, genes in the opposite orientation are placed below the lines.

**Figure 8 marinedrugs-17-00426-f008:**
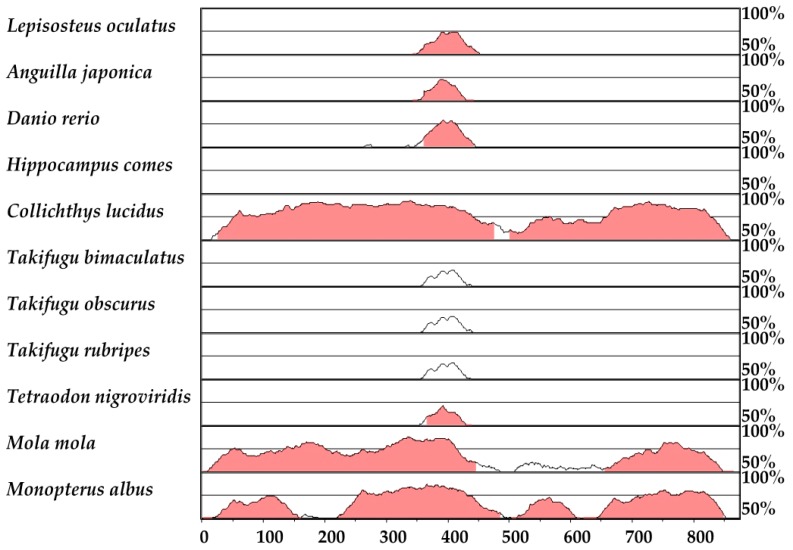
A VISTA plot to compare the HLEB sequences of Acanthopterygii fishes against the reported 873-bp HLEB from three-spined stickleback (*Gasterosteus aculeatus*). Sequence identity along the y-axis, ranging from 50% to 100%, is shown in 100-bp sliding windows across the examined region (x-axis, bp). The pink shadows stand for regions with <100-bp continuous bases at ≥70% identity.

**Table 1 marinedrugs-17-00426-t001:** Summary of the assembling results.

Step	Software	Contig N50 (bp)	Scaffold N50 (bp)	Contig number	Scaffold number	Total length (bp)
Primary assembling	SOAPdenovo	1,999	383,798	1,227,464	462,272	1,167,219,893
Gap filling	krskgf	3,868	375,823	850,121	462,272	1,150,479,312
	Gapclose1.12	5,372	376,296	761,523	462,272	1,154,146,689
	Gapclose1.10	10,215	376,491	624,151	462,272	1,154,798,407
Scaffold extending Filtering	SSPACE ---	10,236 11,468	858,288 1,033,285	608,352 256,649	351,879 41,687	1,228,736,536 1,132,698,062

**Table 2 marinedrugs-17-00426-t002:** A BUSCO assessment of our Japanese eel assembly.

Parameter	Number	Percentage (%)
Complete BUSCOs (C)	3847	83.9
Complete and single-copy BUSCOs (S)	3346	73.0
Complete and duplicated BUSCOs (D)	501	10.9
Fragmented BUSCOs (F)	380	8.3
Missing BUSCOs (M)	357	7.8
Total BUSCO groups searched (n)	4584	---

**Table 3 marinedrugs-17-00426-t003:** Accession numbers for the 27 T-box family members and seven adjacent genes of *tbx4*.

Gene	Species Name	Accession Number
*eomesa*	*Danio rerio*	AAH67719.1
*eomesb*	*D. rerio*	NP_001077044.1
*mgal*	*D. rerio*	XP_021324416.1
*mga*	*D. rerio*	ADA61227.1
*ta*	*D. rerio*	Q07998.1
*tb*	*D. rerio*	XP_001343633.3
*tbr1a*	*D. rerio*	XP_693121.1
*tbr1b*	*D. rerio*	AAG48249.1
*tbx15*	*D. rerio*	AAM54074.1
*tbx16*	*D. rerio*	AAI65213.1
*tbx18*	*D. rerio*	AAI63460.1
*tbx19*	*D. rerio*	XP_003198807.1
*tbx11*	*D. rerio*	XP_017206601.2
*tbx1*	*D. rerio*	Q8AXX2.1
*tbx20*	*D. rerio*	AAF64322.1
*tbx21*	*D. rerio*	NP_001164070.1
*tbx22*	*D. rerio*	ACU00296.1
*tbx2a*	*D. rerio*	AAH68364.1
*tbx2b*	*D. rerio*	Q7ZTU9.4
*tbx3a*	*D. rerio*	NP_001095140.2
*tbx3b*	*D. rerio*	XP_002662050.2
*tbx4*	*D. rerio*	AAI62554.1
*tbx5a*	*D. rerio*	Q9IAK8.2
*tbx5b*	*D. rerio*	ADX53331.1
*tbx6l*	*D. rerio*	P79742.1
*tbx6*	*D. rerio*	Q8JIS6.2
*vegt*	*Fundulus heteroclitus*	JAQ45978.1
*lhx1a*	*D. rerio*	Q90476.1
*brip1*	*Aphyosemion striatum*	SBP21433.1
*bcas3*	*Nothobranchius furzeri*	SBP60348.1
*ppm1da*	*N. furzeri*	SBP60348.1
*appbp2*	*N. furzeri*	SBP60348.1
*usp32*	*N. kuhntae*	SBP60348.1

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
