# Peer review of "Genome Sequencing of the Japanese Eel (Anguilla japonica) for Comparative Genomic Studies on tbx4 and a tbx4 Gene Cluster in Teleost Fishes"

_marinedrugs, 2019, doi:10.3390/md17070426_

Reviewer 1 Report

I thank the authors for their work on the manuscript - the data presentation has been improved and the figures connect to each other better. However, I am still concerned about the main narrative of the paper, which is unfortunately not very strong. The case is made that the assembly of the Japanese eel genome supports the study of the tbx4 gene and locus, and the evolution of fins in teleosts. However, in practice this genome assembly is not used for that purpose, and the biological focus shifts towards the evolution of fin spines, which is interesting from the perspective of the journal but for which the eel is a less relevant footnote.

The eel is an important reference in any study of teleost genome evolution, which is the main reason for sequencing it here. However, the Japanese eel has been sequenced at higher quality already (ref. 74), and it appears that it is that version that is actually used for the comparative study of the tbx4 locus (section 4.4, line 340-342). In addition, it is then not appropriate to compare the assembly presented here to an older assembly only (ref. 16), and claim 'much higher quality' (line 94). Realistically, the assembly presented here is very modest by 2019 standards.

The most interesting points of the paper are in the comparative genomics, figures 6 and 7, with the suggestive inversion of the tbx4 locus in specific groups. Figures 3-5 are somewhat redundant, they mostly show the same data (also used in figure 6).

Section 2.5, line 160 claims two main groups in figure 6. This is actually not what figure 6 shows - in fact, it very nicely shows eels (Elopomorpha) as the earliest branching teleosts, with other lineages diverging later. There is no 'more primitive lineage including Elopomorpha, Osteoglossomorpha' etc. (line 161).

Regarding my earlier comments:

Point 1, on eel genome duplication: the authors include a reference to the preprint I mentioned on a putative eel-specific genome duplication. This paper has now been published by PLoS ONE (last week), however I still find it inconclusive. I would advise against adopting this eel-specific genome duplication as established fact (e.g. abstract line 29, section 2.4 line 129). The analysis presented in figure 4a is insufficient to corroborate this.

Point 2, on duplicate tbx4 loci: Without an eel-specific duplication, I assume the duplicate eel locus in figure 4a originates at the teleost-specific duplication (3R WGD). In which case, other teleost should also have duplicated loci (but perhaps not the genes themselves). You mention zebrafish does not have the locus duplicated, but I think it does, based on simple Ensembl searches for tbx2 and lhx1. The tbx4 gene is lost in one of the zebrafish duplicates, but that is also the case for the eel.

Please carefully check language and spelling. One amusing typo is in the caption for section 2.1: 'gnome survey'!

Author Response

Comments and Suggestions for Authors

I thank the authors for their work on the manuscript - the data presentation has been improved and the figures connect to each other better. However, I am still concerned about the main narrative of the paper, which is unfortunately not very strong. The case is made that the assembly of the Japanese eel genome supports the study of the tbx4 gene and locus, and the evolution of fins in teleosts. However, in practice this genome assembly is not used for that purpose, and the biological focus shifts towards the evolution of fin spines, which is interesting from the perspective of the journal but for which the eel is a less relevant footnote.

Answer: Thanks for your comments. Yes, you are right. Our manuscript is composed of two subtopics, one is about the genome assembly, and the most important part is about the discovery of a special tbx4 gene cluster in teleost fishes. You can understand our arrangement from the title of our paper, which is fit for the special issue of “Genetics of marine organisms associated with human health” in Marine Drugs.

The eel is an important reference in any study of teleost genome evolution, which is the main reason for sequencing it here. However, the Japanese eel has been sequenced at higher quality already (ref. 74), and it appears that it is that version that is actually used for the comparative study of the tbx4 locus (section 4.4, line 340-342). In addition, it is then not appropriate to compare the assembly presented here to an older assembly only (ref. 16), and claim 'much higher quality' (line 94). Realistically, the assembly presented here is very modest by 2019 standards.

Answer: Thanks for your advice. We removed the comparison with an older assembly, which was previously organized under the section 2.1. We sequenced this economically important fish because we are performing an artificial breeding for future mass production in China. Although a high-quality version is available, we are trying to create our own genomic system for a better research or commercial purpose. Since our eel was collected from a different geographic location, it would be valuable to perform biodiversity comparisons between our data and those from other teams. By the way, as we mentioned in the answer to your last question, the most important part of our work is the discovery of a special tbx4 gene cluster in teleost fishes, which is based on a comprehensive comparison of many public available fish genomes.

Specific comments:

1. The most interesting points of the paper are in the comparative genomics, figures 6 and 7, with the suggestive inversion of the tbx4 locus in specific groups. Figures 3-5 are somewhat redundant, they mostly show the same data (also used in figure 6).

Answer: Thanks for your nice advice and comments. We deleted the redundant Figure 4a. However, we kept most of the Figures 3-5, since they describe different topics. In fact, Figure 3 is about the T-box domain in tbox4 genes among various vertebrate species (section 2.3). Figure 4 focuses on the tbx4 NLS regions in different vertebrate species. Figure 5 presents the alignment of the entire TBX4 protein sequences between Japanese eel and zebrafish, which reveals a high conservation among fish species. Localization of the same NLS sequence is marked in a red box for a detailed comparison (upper in Figure 5). Please find more details on lines 116-150.

2. Section 2.5, line 160 claims two main groups in figure 6. This is actually not what figure 6 shows - in fact, it very nicely shows eels (Elopomorpha) as the earliest branching teleosts, with other lineages diverging later. There is no 'more primitive lineage including Elopomorpha, Osteoglossomorpha' etc. (line 161).

Answer: Sorry for the misleading description. This conclusion was based on a previous analysis with human as the outgroup. We changed the outgroup as spotted gar with updated Figure 6. We hence removed the description of two groups, and changed it as follows (on lines 153-154) in accordance with your advice.

It seems that eels (Elopomorpha) appeared as the earliest branching of teleost, with other lineages diverging later.

The earlier comments:

Point 1, on eel genome duplication: the authors include a reference to the preprint I mentioned on a putative eel-specific genome duplication. This paper has now been published by PLoS ONE (last week), however I still find it inconclusive. I would advise against adopting this eel-specific genome duplication as established fact (e.g. abstract line 29, section 2.4 line 129). The analysis presented in figure 4a is insufficient to corroborate this.

Answer: Thanks for your nice advice. Yes, we discarded the description of duplication event (in the Abstract and section 2.4) according to your suggestion.

Point 2, on duplicate tbx4 loci: Without an eel-specific duplication, I assume the duplicate eel locus in figure 4a originates at the teleost-specific duplication (3R WGD). In which case, other teleost should also have duplicated loci (but perhaps not the genes themselves). You mention zebrafish does not have the locus duplicated, but I think it does, based on simple Ensembl searches for tbx2 and lhx1. The tbx4 gene is lost in one of the zebrafish duplicates, but that is also the case for the eel.

Answer: Thanks for your correction and comments. You are right. It seems that the 3R WGD led to the duplication of tbx4 loci. After that, however, the ancestor of the Neopterygii quickly lost one of the redundant tbx4 genes. As mentioned in the answer to your last question, we removed the corresponding description of duplication event (in the Abstract and section 2.4) according to your suggestion.

Please carefully check language and spelling. One amusing typo is in the caption for section 2.1: 'gnome survey'!

Answer: Thanks for your advice. This mistake was fixed on line 79. We also obtained editing help from a colleague, who had worked in the USA for over nine years.

Reviewer 2 Report

Comments to the authors

The manuscript by Chen at al. presents the genome assembly of the Japanese Eel (Anguilla japonica). It is not exactly clear what connection this has to "Marine Drugs", the manuscript certainly does not do a very good job establishing it, but be that as it may. Based on the presence of text highlighted in yellow, this appears to be a resubmission from a previous submission that I have not reviewed. It is also readily apparent what some of the points the previous reviews focused on were, as well as that they have not been entirely properly dealt with. But even aside from those, there are several issues with the paper as sent to me that prevent it from being published in its current form.

Specific comments

1. First, the manuscript is horribly written and should not be under any circumstances published without a thorough rewriting by someone who knows English sufficiently well to carry out that task.

2. However, merely fixing the grammar is not going to fix the more general problem having to do with the fact that it is also not organized properly. Sections are just thrown together without much connection between them, and the introduction does a very poor job of introducing the species, why it is important to sequence it, and why its genome was analyzed the way it was.

3. The assembly itself looks quite suspicious. Achieving an N50 of nearly a megabase is quite nice on the surface. However, the k-mer analysis suggests a genome size of nearly 200 Mbp less than the actual assembly. This raises the obvious concern that high rates of heterozygosity are driving an artificial expansion of the assembly. Moreover, it appears that this concern was already raised previously, because in one of the highlighted in yellow paragraphs we see the sentence \It seems that the genome of the Japanese eel is of high heterozygosity (Figure 1A)\. However, Figure 1 has no panel \A" in it, nor does it have anything in it that relates to heterozygosity.

4. Despite the likely inflated assembly size, the BUSCO results indicate that nearly 10% of the BUSCO set genes are missing, which is far from optimal.

5. There is presumably more to be learned about eel biology from the genome sequence than just focusing onTbx4, and it would be nice to explore that in a revised version of the paper.

Author Response

Comments and Suggestions for Authors

Comments to the authors

The manuscript by Chen at al. presents the genome assembly of the Japanese Eel (Anguilla japonica). It is not exactly clear what connection this has to "Marine Drugs", the manuscript certainly does not do a very good job establishing it, but be that as it may. Based on the presence of text highlighted in yellow, this appears to be a resubmission from a previous submission that I have not reviewed. It is also readily apparent what some of the points the previous reviews focused on were, as well as that they have not been entirely properly dealt with. But even aside from those, there are several issues with the paper as sent to me that prevent it from being published in its current form.

Specific comments

1. First, the manuscript is horribly written and should not be under any circumstances published without a thorough rewriting by someone who knows English sufficiently well to carry out that task.

Answer: Thanks for your advice and comments. Yes, we obtained editing help from a colleague, who had worked in the USA for over nine years. A thorough revision of the manuscript, especially for the Abstract and Introduction, was realized for your consideration.

2. However, merely fixing the grammar is not going to fix the more general problem having to do with the fact that it is also not organized properly. Sections are just thrown together without much connection between them, and the introduction does a very poor job of introducing the species, why it is important to sequence it, and why its genome was analyzed the way it was.

Answer: Our manuscript is composed of two subtopics, one is about the genome assembly, and the most important part is about the discovery of a special tbx4 gene cluster in teleost fishes. You can understand our arrangement from the title of our paper, which is fit for the special issue of “Genetics of marine organisms associated with human health” in Marine Drugs. As mentioned in the answer to the last question, we obtained editing help from a colleague, who had worked in the USA for over nine years.

3. The assembly itself looks quite suspicious. Achieving an N50 of nearly a megabase is quite nice on the surface. However, the k-mer analysis suggests a genome size of nearly 200 Mbp less than the actual assembly. This raises the obvious concern that high rates of heterozygosity are driving an artificial expansion of the assembly. Moreover, it appears that this concern was already raised previously, because in one of the highlighted in yellow paragraphs we see the sentence \It seems that the genome of the Japanese eel is of high heterozygosity (Figure 1A)\. However, Figure 1 has no panel \A" in it, nor does it have anything in it that relates to heterozygosity.

Answer: Sorry for the mistake. The corresponding description under section 2.1 was removed. In fact, we planned to include two panels in accordance with the opinion from a reviewer. However, due to our limitations to analyze the heterozygosity of the Japanese eel genome, we removed one updated panel but forgot to correct the revised information (Figure 1A should be Figure 1).

4. Despite the likely inflated assembly size, the BUSCO results indicate that nearly 10% of the BUSCO set genes are missing, which is far from optimal.

Answer: Yes, you are right. It seems that the Japanese eel genome was quite complex. We are planning to perform PacBio and HiC sequencing to improve our present genome assembly.

5. There is presumably more to be learned about eel biology from the genome sequence than just focusing onTbx4, and it would be nice to explore that in a revised version of the paper.

Answer: Thanks for your suggestion. We would like to explore more biomedical issues on the Japanese eel, once we establish a chromosome-level genome assembly with assistance of PacBio and HiC sequencing. Due to time limit, we have to focus on the discovery of a special tbx4 gene cluster in teleost fishes, which is fit for the special issue of “Genetics of marine organisms associated with human health” in Marine Drugs.

Reviewer 3 Report

I feel much of the narrative in the abstract and Introduction are confusing and somewhat unnecessary. It is not until the last sentence of the abstract and Introduction that the authors articulate that the tbx cluster inversion event may be correlated with fin spines (and indirectly toxin storage). Would it not make more sense to focus on this funding, rather than provide an extremely broad overview of the role of tbx in appendage evolution?

Similarly, it is somewhat unclear to me what is the primary focus of the paper - is this the report of a new genome? Or, alternatively, is this a report focusing on the tbx gene family? It appears to be both - however, it is rather unclear. Did the authors perform genome sequencing with the sole focus of evaluating tbx genes? If so, why not just clone these family members without whole genome sequencing?

The authors state: "It seems that one tbx4 copy was absent; the other one contains a non-synonymous mutation in an important site of the nuclear localization sequence, which was

considered to be correlated with the pelvic fin development, and interestingly its adjacent brip1 gene was also lost." Since this is the conclusion, can the authors provide a conclusion to this statement? Do they feel fin development is altered specifically due to this nonsynonymous mutation? 

Author Response

Comments and Suggestions for Authors

I feel much of the narrative in the abstract and Introduction are confusing and somewhat unnecessary. It is not until the last sentence of the abstract and Introduction that the authors articulate that the tbx cluster inversion event may be correlated with fin spines (and indirectly toxin storage). Would it not make more sense to focus on this funding, rather than provide an extremely broad overview of the role of tbx in appendage evolution?

Answer: Thanks for your nice advice. However, because short of supportive evidence, we have to only mention the inversion of tbx4 cluster in some fish species. In fact, our manuscript is composed of two subtopics, one is about the genome assembly, and another part is about the discovery of a special tbx4 gene cluster in teleost fishes. You can understand our arrangement from the title of our paper, which is fit for the special issue of “Genetics of marine organisms associated with human health” in Marine Drugs.

Similarly, it is somewhat unclear to me what is the primary focus of the paper - is this the report of a new genome? Or, alternatively, is this a report focusing on the tbx gene family? It appears to be both - however, it is rather unclear. Did the authors perform genome sequencing with the sole focus of evaluating tbx genes? If so, why not just clone these family members without whole genome sequencing?

Answer: We have to combine the genome sequencing and a primary analysis of tbx4 gene cluster for the special issue in Marine Drugs. In fact, as we described in the main text (for example, on lines 19-22 of the Abstract), we originally aimed at examination of the correlation between natural pelvic-fin-loss and tbx4 gene in more fish species, although we have confirmed the pelvic-fin-loss in tbx4 knock-out zebrafish (Nature, 2016, 540:395-399). However, it seems that this issue is very complicated, since “we observed the complete exon structures of tbx4 in several pelvic-fin-loss species, such as Ocean sunfish (Mola mola) and ricefield eel (Monopterus albus)” (lines 26-28). We therefore had to narrow the current research topic into the interesting inversion of tbx4 cluster in some fish species, since it is somehow correlated with the evolutionary development of toxic fin spines (lines 35-37). Because the fin spines in teleost fishes are usually venoms for storage of various toxins, this tbx4 gene cluster will benefit for genetic engineering development of toxin related marine drugs (lines 37-38 in the Abstract).

The authors state: "It seems that one tbx4 copy was absent; the other one contains a non-synonymous mutation in an important site of the nuclear localization sequence, which was

considered to be correlated with the pelvic fin development, and interestingly its adjacent brip1 gene was also lost." Since this is the conclusion, can the authors provide a conclusion to this statement? Do they feel fin development is altered specifically due to this nonsynonymous mutation?

Answer: Thanks for your advice and questions. Based on another reviewer’s advice, we removed the first part of this sentence. However, we kept the description of the non-synonymous mutation on lines 131-133. Indeed, using a TALEN-induced tbx4 knockout allele, Don et al. (2016) [ref. 20 in our revised manuscript] confirmed that mutations within the Tbx4 NLS (A78V; G79A) are sufficient to disrupt pelvic fin development in zebrafish. We mentioned this study in Discussion (on lines 226-227), and updated this information in the Conclusions as follows (on lines 352-354).

It seems that its tbx4 contains a non-synonymous mutation in an important site of the nuclear localization sequence, which was considered to be correlated with the pelvic fin development; interestingly, its adjacent brip1 gene was also lost.

Round  2

Reviewer 1 Report

The manuscript's writing has been improved, minor errors have been fixed, and any confusion on genome duplication events has been removed. The issues with the narrative and the necessity of the genome assembly (or rather, lack thereof) are intrisic to the paper, I'm afraid.

Author Response

Comments and Suggestions for Authors

The manuscript's writing has been improved, minor errors have been fixed, and any confusion on genome duplication events has been removed. The issues with the narrative and the necessity of the genome assembly (or rather, lack thereof) are intrisic to the paper, I'm afraid.

Answer: Thanks for your positive comments. In the past week, we obtained an editing service from a professional company, LetPub (No. 190710J05), to improve our overall writing. Please find the trackable changes in our revised manuscript.

For the necessity of the genome assembly, we think that it is at least useful for (1) characterization of the tbx4 gene in the Japanese eel, another important fish without pelvic fins; (2) fitting this special issue in Marine Drugs with phylogenetic determination of the special tbx4 gene cluster, which is potentially correlated with the evolutionary development of toxic fin spines, and therefore facilitates the genetic engineering of toxic-related marine drugs; and (3) public availability of genomic data for the sample from a different geographic location, which may provide helpful SNPs for further artificial breeding.

Reviewer 2 Report

The authors have not really addressed my concerns. The language has improved only slightly, there are still plenty of sentences such as ``it was a common thought that tetrapod forelimbs and hindlimbs are the homologies of fish pectoral and pelvic fins, respectively'' throughout the text. Organization is still poor. The heterozygosity issue was just brushed aside rather than explored and characterized, something that is fairly straightforward to do directly from the sequence. Same for my other comments -- they are acknowledged, but that is not the same as actually addressing them.

Author Response

Comments and Suggestions for Authors

The authors have not really addressed my concerns. The language has improved only slightly, there are still plenty of sentences such as ``it was a common thought that tetrapod forelimbs and hindlimbs are the homologies of fish pectoral and pelvic fins, respectively'' throughout the text. Organization is still poor. The heterozygosity issue was just brushed aside rather than explored and characterized, something that is fairly straightforward to do directly from the sequence. Same for my other comments -- they are acknowledged, but that is not the same as actually addressing them.

Answer: Thanks for your instructive comments. Yes, in the past week, we obtained an editing service from a professional company LetPub (No. 190710J05) to improve our overall writing. Please find the trackable changes in our revised manuscript.

Meanwhile, we performed additional filtering to remove many heterozygous redundant scaffolds manually. In fact, these heterozygous scaffolds were sequenced with lower depths (<< span=""> 40Ă—; ~1/4 of the average sequencing depth). We hence generated a final 1.13-Gb genome assembly with improved contig N50 and Scaffold N50 values. Please find more details in the revised Section 2.1. (lines 91-94), Table 1 (page 3), and Section 4.2. (lines 293-296).

You are right. Our present genome assembly and corresponding BUSCO results are far from optimal, although missing of 7.8% BUSCOs is somehow at a mid-level for genome assembling with the second generation of sequencing technology. Here, we provide additional genomic data for future diversity analysis and artificial breeding. The major achievement of our present work is to reveal a potential correlation between the tbx4 cluster and the evolutionary development of toxic fin spines. We are going to perform PacBio and HiC sequencing to obtain a chromosome-level genome assembly.

By the way, we removed the sentence about “high heterozygosity” under the Section 2.1., which was mentioned in our previous responses.

Reviewer 3 Report

Although the authors provided responses to my queries, I still feel that the presentation of the material is rather vague and unclear at times. I feel the authors must provide an additional round of proofreading to get the manuscript to a level of clarity which is necessary for this paper to have an appropriate level of impact. 

Author Response

Comments and Suggestions for Authors

Although the authors provided responses to my queries, I still feel that the presentation of the material is rather vague and unclear at times. I feel the authors must provide an additional round of proofreading to get the manuscript to a level of clarity which is necessary for this paper to have an appropriate level of impact.

Answer: Thanks for your instructive comment. Yes, we obtained an extensive editing service from a profession company LetPub (No. 190710J05) to improve our overall writing. Please find the trackable changes in our revised manuscript.

Round  3

Reviewer 2 Report

There are still some typos and linguistic errors, but the authors have mostly addressed my concerns. 

This manuscript is a resubmission of an earlier submission. The following is a list of the peer review reports and author responses from that submission.

Round  1

Reviewer 1 Report

This paper presents three perspectives on teleost fish evolution:

1) A genome assembly of the Japanese eel;

2) An investigation of the tbx4 locus, which might be involved in pelvic fin loss;

3) The suggestion that an inversion at this locus is correlatied with fin spine evolution.

The topics themselves are highly interesting, and the Japanese eel is an interesting outgroup for many aspects of teleost evolution and development. However, there is insufficient coherence between the separate perspectives. Most importantly, the Japanese eel is not included in the analysis related to fin spine development (figs 8 & 9). Even if it doesn't have them, it should be included as an outgroup here to provide a rationale for sequencing the genome (and a narrative for the paper).

In general, each analysis (figure) appears to select its own set of reference genomes. This, at least, can be easily fixed, and would significantly increase the overall coherence. Specifically, I miss seahorse after figure 4 (it is explicitly mentioned as a reason for conducting this study, p2 line 56); eel in figures 7-9; spotted gar (Lepisosteus aculatus), a very relevant outgroup, in fig 6; and zebrafish, the best established teleost model in figs 8 & 9. 

Specific comments:

1. A. japonica is not tetraploid, as mention in the abstract (l 21) and p5, line 133. There very likely has been no Anguilliformes-specific whole genome duplication event (l 28). (There is a preprint claiming this, https://www.biorxiv.org/content/10.1101/232918v1, but the evidence is inconclusive at best.)

2. The tetraploidy mentioned in section 2.4 is ancestral to all teleosts (also according to the paper referenced here), and by now (300 million years later) is no longer referred to as such. Rather, this is an ancient whole-genome duplication event. After this WGD, each branch of teleost has lost different duplicated genes. In figure 4a, this is shown for A. japonica. However, D. rerio and H. comes should also have duplicate tbx4 loci - which probably have lost the tbx4 gene itself (as has eel copy 2), but can be identified based on conserved synteny. These loci should be included in this figure.

3. The genome assembly has been performed using suitable methodology. Some claims,however, need further investigation, especially on the genome size. At 1.23 Gbp, it is relatively large, and larger than most other eel genome assemblies. Could this be caused by the high heterozygosity of this genome (figure 1)? This potentially confuses De Bruijn-graph assemblers. This could be checked by assessing the coverage depth per contig. This issue has been discussed in the recent European eel genome paper (https://www.nature.com/articles/s41598-017-07650-6).

4. The completeness of the genome should be checked by BUSCO analysis.

5. The genome assembly should be made publicly available.

6. Other (Japanese) eel genome efforts should be clearly acknowledged. Especially the 2018 chromosome-anchored version (https://journals.plos.org/plosone/article?id=10.1371/journal.pone.0201784) which is more advanced than the one presented here. This is even acknowledged in the methods (p12, line 345), as this assembly has been used for some of the analyses.

7. With regards to the A. japonica tbx4 gene, as the exon structure is highlighted across teleosts (figure 2) it would be appropriate to analyze this for the eel as well.

8. Figure 7: this disagreement with established phylogeny is based on low bootstrap support (figure 6), and therefore not conclusive. In any case, in a paper on the eel genome, I would expect the emphasis to be placed on the disagreement of the placement of Elopomorpha (+ Osteoglossomorpha) with established phylogeny, which has them as the earliest diverging teleosts (and not members of the cyprinid branch). I assume this is a case of long branch attraction paired to a less suitable outgroup (H. sapiens). It would be very appropriate to include spotted gar in this analysis.

9. Figure 8. This figure does not show exons and introns, as the legend states. Rather, it appears to display entire gene clusters, colour-coded as in figure 6. This should be made clear. In that case, figure 8 and 6 are inconsistent: the order of genes is not the same for both I. punctatus and O. niloticus, as well as (I assume) for D. rerio/D. dracula. Based on figure 8, there are no cluster inversions but individual genes moving around.

Tiny comments:

p2 l45: Fingerlings - I do not know this term applied to eel life stages. Probably glass eels?

p2 l52: Tetrapods have not retained the swimming abilities of their fish ancestors. You probably mean the earliest tetrapods?

p2 l63: Merged by duplication -> diverged by duplication.

p3 l108: I would suggest replacing 'Leptocard' (= class Leptocardii) with Cephalochordata.